# Genotypic and phenotypic prevalence of *Nocardia* species in Iran: First systematic review and meta-analysis of data accumulated over years 1992-2021

**Mohammad Hashemzadeh**[1,2,‡], **Aram Asareh Zadegan Dezfuli**[2], **Azar Dokht Khosravi**[1,2,*], **Mohammad Savari**[1,2], **Fatemeh Jahangirimehr**[3]

1 Infectious and Tropical Diseases Research Center, Health Research Institute, Ahvaz Jundishapur University of Medical Sciences, Ahvaz, Iran, 2 Department of Microbiology, Faculty of Medicine, Ahvaz Jundishapur University of Medical Sciences, Ahvaz, Iran, 3 Pain Research Center, Ahvaz Jundishapur University of Medical Sciences, Ahvaz, Iran

☯ These authors contributed equally to this work.
‡ MH is Senior Author.
* azarkhosravi69@gmail.com

**Data Availability Statement:** All relevant data are within the manuscript.

## Abstract

### Background

*Nocardia* species belong to the aerobic actinomycetes group of bacteria which are gram-positive and partially acid-fast Bacilli. These bacteria may sometimes be associated with nosocomial infections. *Nocardia* diseases are not required to be reported to public health authorities in Iran. Hence, the present study was designed to determine the prevalence of human *Nocardia* spp. in Iran by using a systematic review and meta-analysis according to the preferred reporting items for systematic reviews and meta-Analyses statement.

### Methods

The data of the prevalence of *Nocardia* species were collected from databases such as Embase, PubMed/MEDLINE via Ovid, Web of Science, Scopus and Google Scholar as well as national Iranian databases, including SID, Magiran. Analyses were conducted by STATA 14.0.

### Results

The meta-analyses showed that the proportion of *Nocardia* spp. in Iranian studies varied from 1.71(1.17, 2.24) to 0.46(0.09, 0.83). *N. asteroides* (21% [95% CI 1.17, 2.24]), *N. cyriacigeorgica* (17% [95% CI 0.99, 1.77]), *N. facanica* (10% [95% CI 0.75, 1.00]) were considered to be common causative agents.

### Conclusions

Our study presents that despite the fact that *Nocardia* spp. are normally are saprophytic organisms, are currently accounts as emerging pathogens due to an increase in

**Funding:** This work is part of a research project which was approved in Infectious and Tropical Diseases Research Center, Ahvaz Jundishapur University of Medical Sciences, Ahvaz, Iran, and was supported by a grant (No.: OG-9829) from research affairs of the university.

**Competing interests:** The authors have declared that no competing interests exist.

immunocompromised patients among Iranian populations. Considering our results, the establishment of advanced diagnostic facilities for the rapid detection of *Nocardia* infections are required for optimal therapeutic strategies of *Nocardia* spp. in Iran. Our findings could help the programmatic management of the disease within the context of *Nocardia* control programs.

## Introduction

*Nocardia* species (spp.) belong to the aerobic actinomycetes group of bacteria which are gram-positive and partially acid-fast bacilli (AFB) [1]. These bacteria are saprophytic and are found in soil and water, however, they may sometimes be associated with nosocomial infections [2]. More than 40 of the approximately 86 *Nocardia* spp. characterized, have been involved in human infections and include *Nocardia asteroides complex* (more than 50% human cases), *N. brasiliensis*, *N. abscessus*, *N. cyriacigeorgica*, *N. farcinica*, *N. nova*, *N. transvalensis complex*, *N. novacomplex*, *N. pseudobrasiliensis*, and the recently described spp. include *N. veterana*, *N. paucivorans*, *N. elegans*, *N. wallacei* and *N. blacklockiae* of the *N. transvalensis complex* [3]. Human *Nocardia* infection can be airborne via inhalation of dust particles (pulmonary nocardiosis-pneumonia, lung abscess) or skin infections (cutaneous nocardiosis-cellulitis, ulcers), and the infection can then disseminate to the brain, kidneys, joints, heart, eyes, and bones [4]. So far, person to person transmission is not documented [3]. Pulmonary nocardiosis is a low but severe opportunistic infectious disease and mainly affects patients with compromised cell-mediated immunity, such as those experiencing long-term steroid use, patients with acquired immunodeficiency syndrome (AIDS), or recipients of organ transplantation [5]. However, early diagnosis of pulmonary nocardiosis may be difficult because signs and symptoms in the affected patients are nonspecific and very similar to tuberculosis (TB) [6].

Co-infection with *Nocardia* spp. in patients suffering from mycobacterial lung infection, highlights the importance of laboratory diagnosis that may facilitate better patient management. The diagnosis of nocardiosis is currently based on direct examination and conventional culture, while serology is usually not useful [7]. The molecular methodologies which have provided precise identification of *Nocardia* spp. are important for studies of clinical and epidemiological investigations [8]. Unlike tuberculosis, *Nocardia* diseases are not required to be reported to public health authorities in Iran, and therefore, precise incidence and prevalence data are not available. In order to estimate what the true burden of *Nocardia* human infections is global, a meta-analysis was performed to review all studies related to the epidemiology of the *Nocardia* causative agents. Hence, the present study was designed to determine the prevalence of human *Nocardia* spp. In Iran by using a systematic review and meta-analysis according to the preferred reporting items for systematic reviews and meta-Analyses statement.

## Methods

### Ethics approval and consent to participate

The initial proposal of the work was approved by the Institutional Review Board (IRB) and Ethics Committee of the Ahvaz Jundishapur University of Medical Sciences, Iran, and necessary permission was granted for the work (IR.AJUMS.REC.1398.538).

## Literature search

A systematic review of available literature was searched using the electronic database such as: Embase, PubMed/MEDLINE via Ovid, Web of Science, Scopus and Google Scholar as well as national Iranian databases, including SID, Magiran, with medical subject headings (MeSH) terms and a proper use of keywords. The search strategy was as follows: " *Nocardia* ", "nocardiosis", "*Nocardia* and human infection", "actinomycete" and "Iran". Original articles on *nocardia* and a time filter (from August 1992 to January 2021) applied including Persian and English articles were considered. Likewise, the full texts of potentially relevant articles were assessed for eligibility independently and in duplicate by two investigators. In addition to articles published in English, we also looked for relevant articles in Persian.

## Inclusion and exclusion criteria

After the search results were merged into Endnote (X7; Thomson Reuters), the resultant was de-duplicated and screened by applying a Rayyan Qatar Computing Research Institute online application. Criteria considered for inclusion were cross-sectional surveys assessing the frequency or prevalence of *Nocardia* spp. in Iran. Based on the research protocol and the eligibility criteria, the titles and abstracts were separately retrieved by two independent researchers. Following the elimination of repetitive studies, the full text of the papers in terms of eligibility criteria and the required extracted necessary information were studied. Disagreements between the two researchers were resolved by a consensus method. The final data extracted from the search results included corresponding author, year, place, research design, sample size, location, study period, individual *Nocardia* spp., and detection method. The exclusion criteria were including the papers with the following features: review articles, meta-analyses or systematic reviews, case reports and letter to editor studies, congress abstracts, and the duplication papers, as well as articles in languages other than English or Persian and those available only in abstract form. To evaluate the eligibility of the articles with inadequate information, we made a contact with the corresponding author. Culture as well as biochemical and molecular testes were the standard methods for detection.

To conduct phenotypic methods, the paraffin baiting technique was used and samples were cultured on various cyclohexamide-containing agars (i.e. blood, nutrient, and Sabouraud Dextrose) and were incubated at 35°C. Kinyoun acid-fast stain and Gram stain were used for initial investigation of colonies grown on culture media. The partially acid-fast and Gram-stained organisms showing that colonies bore a resemblance to the genus *Nocardia*. Stereomicroscopy was employed to assess the morphology of colonies. Numerous biochemical tests employed on the grown colonies in the present work were as follows: decomposition of L-tyrosine, growth in lysozyme broth and also at 45°C, hydrolysis of casein, esculin, gelatin, urea, xanthine, and hypoxanthine, utilization of citrate, and production of nitrate reductase, as well as acid production of sorbitol, rhamnose, glucose, L-arabinose, D-xylose, galactose, mannitol, lactose, maltose, sucrose, raffinose, and salicin.

## Data extraction

Two reviewers independently extracted the data from eligible studies. According to inclusion and exclusion criteria, all collected data from the selected studies were tabulated as follows: (1) First author, (2) publication date, (3) enrollment time, (4) province of study, (5) all patients included in study, and (6) prevalence of *Nocardia* human infections. Two authors extracted data from involved studies independently. Inconsistency between the reviewers was resolved through discussion.

## Quality assessment

The quality of papers was evaluated using the Strengthening the Reporting of Observational studies in epidemiology (PRISMA) checklist and the guidelines of the Cochrane Handbook for Systematic Reviews and Interventions [9]. This checklist has 8 parts which covers different sections of reports. If necessary, the authors were contacted for further information.

## Statistical analysis

In this study, the prevalence of *Nocardia* in the country was collected and then the variance of each study was determined by Double arcsine conversion method. The point estimates of effect size, the prevalence of *Nocardia* spp., and its 95% confidence interval (95% CI) were estimated for each study. Random effects models were used, taking into account the possibility of heterogeneity between studies, which was tested with the Cochran's Q- and the I2 statistics. In order to assess possible publication bias, Egger weighted regression methods were used. Value of $P < 0.05$ was considered indicative of statistically significant publication bias. Analyses were conducted by STATA 14.0 (StataCorp, College Station, TX, US).

## Results

### Characteristics of the included studies

A total of 93 articles were obtained by a literature search with a combination of keywords from the databases as shown in Fig 1. In secondary screening and after duplication, 18 articles were identified and were removed due to the irrelevant titles. Then based on the abstract evaluation, 55 articles were excluded (3 review articles, 27 case reports, 8 letters to the editor, and 17 were related to non-clinical *Nocardia* specimens). So, according to quality assessment criteria and inclusion/exclusion criteria data, a remaining 20 most-related articles were included in present study [10–29]. Among 20 articles involving a total of 338 *Nocardia* isolates, the prevalence of *Nocardia* spp. were recorded. The articles were published between years 1994 to 2021. The characteristics of the selected articles are summarized in Table 1.

### The prevalence of different *Nocardia* spp.

In total, 338 different *Nocardia* spp. were identified in the studied Iranian articles, with the varied proportion from 1.71(1.17, 2.24) to 0.46 (0.09, 0.83) using 90% confidence interval Table 2. *N. asteroides* (21% [1.17, 2.24]), *N. cyriacigeorgica* (17% [95% CI 0.99, 1.77]), *N. facanica* (10% [0.75, 1.00]) were considered to be the most common causative agents, while, *N. coubleae* (0/0011% [0.09, 0.80]), *N. cummidelens* (0/0011% [0.09, 0.80]), *N. neocaledoniensis* (0/0011% [0.24, 1.84]) and *N. ignorata* (0/0011% [0.09, 0.80]) isolates were considered as the uncommon causative agents mentioned in only one study. It is necessary to mention that the causative agents were not identified to the spp. level in 12% of cases [01.67, 2.86]). Fig 2 shows the forest plot of meta-analysis of *Nocardia* prevalence. Some evidence for publication bias was observed in Fig 3.

### The prevalence of different *Nocardia* spp. in provinces of Iran

Out of these 20 articles, 15 were belong to reports from Tehran, center of Iran. The rest were as follows: 5 from southwest of Iran (4 of them from Khuzestan and 1 from Kermanshah provinces), Isfahan, Yazd, Central (Arak), and Golestan provinces one report each. Fig 4 shows the distribution of *Nocardia* spp. in different parts of Iran. The *Nocardia* isolation in the central provinces of Iran demonstrated apparent characterization, as from the central province (Tehran) to the southwest province (Khuzestan, Kermanshah) the *Nocardia* isolation rate was

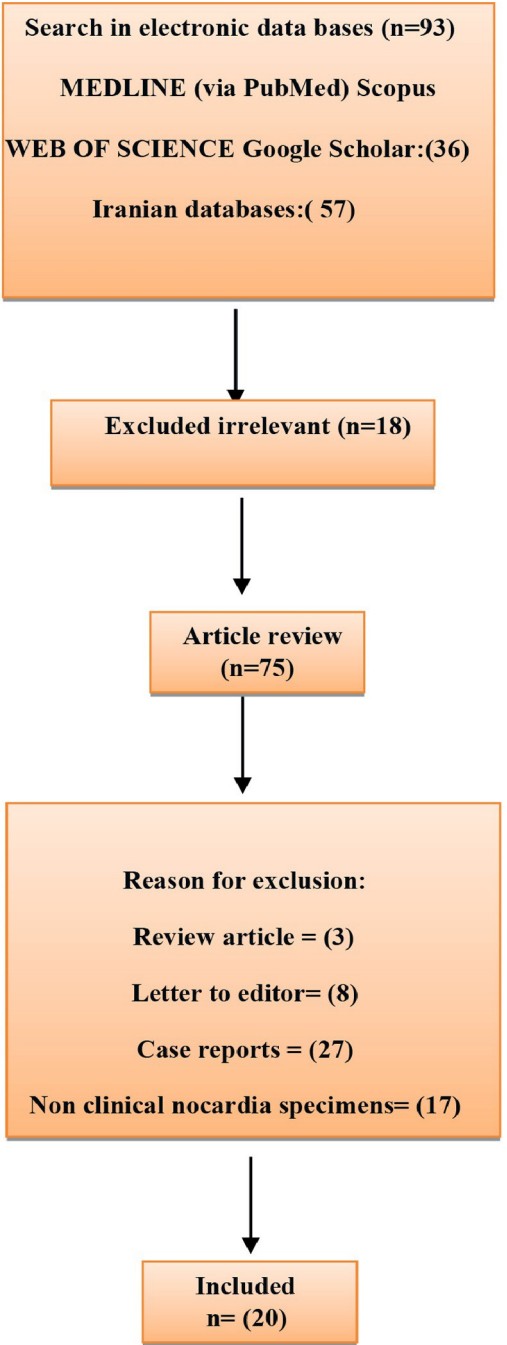

**Fig 1. Flow diagram of literature search.**

increased. The most frequent isolated *Nocardia* spp. in central provinces were *N. otitidisca-viarum.caviae* and *N. cyriacigeorgica*, while *N. farcinica* and *N. wallacei* were the most typical isolated species (Table 3).

**Table 1. Characteristics of studies involved in the current systematic review and meta-analysis.**

| Articles | Authors | Time of study | Publication date | City | Methods | No. of specimens | Nocardia isolation |
|---|---|---|---|---|---|---|---|
| 1 | Bafghi | 2008–2015 | 2016 | Tehran | PCR—Conventional tests | 789 | 27 |
| 2 | Eshraghi | 1998–1999 | 2001 | Tehran | Conventional tests | 102 | 1 |
| 3 | Taheri javan | 2013 | 2015 | Yazd | PCR-Conventional tests | 100 | 4 |
| 4 | Heidarzadeh | 2010–2011 | 2011 | Tehran | PCR-Conventional tests | 180 | 19 |
| 5 | Zaker Bostanabad | 2012–2013 | 2014 | Ahvaz/Tehran | Conventional tests | 90 | 6 |
| 6 | Faghri | 2004 | 2007 | Isfahan | Conventional tests | 200 | 8 |
| 7 | Bafghi | 2011–2013 | 2015 | Tehran | Conventional tests | 517 | 7 |
| 8 | Ekrami | 2011–2012 | 2014 | Ahvaz | PCR-Conventional tests | 189 | 2 |
| 9 | Bafghi | 2012 | 2014 | Tehran | Conventional tests | 250 | 2 |
| 10 | Abtahi | 2000 | 2003 | Arak | Conventional tests | 600 | 26 |
| 11 | Eshraghi | 2003 | 2004 | Tehran | Conventional tests | 150 | 2 |
| 12 | Hashemi-Shahraki | 2009–2015 | 2015 | Multi- regions* | PCR | 789 | 127 |
| 13 | Kordbache | 1990–1992 | 1994 | Tehran | Conventional tests | 170 | 5 |
| 14 | Zaker Bostanabad | 2009–2012 | 2014 | Tehran-Ahvaz | PCR—Conventional tests | 160 | 46 |
| 15 | Famili | 2011–2012 | 2015 | Tehran | PCR-Conventional tests | 116 | 7 |
| 16 | Bolourchi | 2017–2018 | 2019 | Tehran | Real-PCR-Conventional tests | 25 | 3 |
| 17 | Gharebaghi | 2018 | 2019 | Tehran | PCR—Conventional tests | 200 | 29 |
| 18 | Larijanian | 2011–2015 | 2018 | Tehran | Conventional tests | 465 | 9 |
| 19 | Rahdar | 2018–2019 | 2019 | Tehran | PCR—Conventional tests | 29 | 3 |
| 20 | Azadi | 2018–2019 | 2020 | Arak | PCR—Conventional tests | 79 | 5 |

*Tehran-Khuzestan-Golestan-Kermanshah-Isfahan.

## The prevalence of *Nocardia* spp.in clinical specimens

The distribution of pulmonary nocardiasis sites shown in Table 4. Among the patients with pulmonary Nocardia infection 139 out of 259 (53%) with bronchoalveolar lavage (BAL) and 120(46%) was sputum. Among the patients with extra pulmonary specimen, 4(2%) with wound, 26 (32%) abscess, 8 (27%) blood, one specimen was pleural and 30 (26%) skin. The distribution of extra pulmonary sites shown in Table 5.

## Discussion

Due to the low probability of transmission among people, *Nocardia* infection was not taken into account as a public health problem, and its incidence still remains unknown [30]. Literature review have revealed that the number of clinical isolations of *Nocardia* infection is escalating globally, possibly in relation to the increased number of immunocompromised patients and improved laboratory techniques for nocardiosis detection [31, 32]. To our knowledge, this is the first study in Iran which focuses on the incidence of *Nocardia* spp. among Iranian population.

The analysis has revealed a relatively high prevalence of *Nocardia* spp. among Iranian patients. The overall prevalence of this species was also greater when the study was performed after the year 2000 with the rate of 54% compared to the rate before year 2000 as 38%. The higher number of reports of *Nocardia* spp. appears to be due to the significance of the disease identified by microbiologists and physicians, advancement of laboratory facilities, and increasing the incidence of immunocompromised hosts, which gives rise to a growth in *Nocardia* spp. in the entire population. In general, the relatively high incidence (0.49%) of *Nocardia* spp. in our country may have an adverse impact on public health. Despite the implementing of

**Table 2.** *Nocardia* spp. distribution among Iranian studies.

| Nocardia spp. | N. of studies | N/% | Prevalence of *nocardia* (95% CI*) | Heterogeneity test I² (%) | Heterogeneity test P-Value | Eggers test t | Eggers test p-value |
|---|---|---|---|---|---|---|---|
| *N. asteroides* | 9 | 74/(21%) | 1.71(1.17, 2.24) | 92.8 | <0.001 | 1.34 | 0.228 |
| *Nocardia.spp* | 8 | 42/(12%) | 2.27(1.67, 2.86) | 78.7 | <0.001 | 0.78 | 0.470 |
| *N. cyriacigeorgica* | 6 | 60/(17%) | 1.38(0.99, 1.77) | 86.0 | <0.001 | 1.84 | 0.139 |
| *N. farcinica* | 6 | 41/(12%) | 0.87(0.75, 1.00) | 0.0 | 0.562 | -0.06 | 0.956 |
| *N. otitidiscaviarum. caviae* | 5 | 40/(11%) | 0.66(0.49, 0.82) | 30.0 | 0.222 | 0.58 | 0.601 |
| *N. nova* | 5 | 7/(0/02) | 0.38(0.25, 0.50) | 25.7 | 0.250 | 7.11 | 0.006 |
| *N. wallacei* | 3 | 14/(0/04) | 0.70(0.35, 1.05) | 71.2 | 0.031 | 3.12 | 0.198 |
| *N. arthritidis* | 3 | 3/(0/008) | 0.28(0.14, 0.42) | 0.0 | 0.442 | 1.86 | 0.145 |
| *N. a.complex* | 2 | 5/(0/014) | 1.22(0.80, 1.64) | 65.0 | 0.091 | - | - |
| *N. carnea* | 2 | 6/(0/017) | 0.39(0.24, 0.54) | 39.9 | 0.197 | - | - |
| *N. kruczakiae* | 2 | 2/(0/008) | 0.25(0.10, 0.40) | 0.0 | 0.411 | - | - |
| *N. abscessus* | 2 | 33/(0/09) | 0.59(0.16, 1.03) | 85.5 | 0.009 | - | - |
| *N. veterana* | 2 | 2/(0/008) | 0.25(0.10, 0.40) | 0.0 | 0.411 | - | - |
| *N. brasiliensis* | 2 | 2/(0/008) | 0.53(0.22, 0.84) | 0.0 | 0.351 | | |
| *N. transvalensis* | 1 | 1/(0/0011) | 0.46(0.09, 0.83) | - | - | - | - |
| *N. coubleae* | 1 | 1/(0/0011) | 0.44(0.09, 0.80) | - | - | - | - |
| *N. cummidelens* | 1 | 1/(0/0011) | 0.44(0.09, 0.80) | - | - | - | - |
| *N. ignorata* | 1 | 1/(0/0011) | 0.44(0.09, 0.80) | - | - | - | - |
| *N. mexicana* | 1 | 2/(0/008) | 1.40(0.60, 2.20) | - | - | - | - |
| *N. neocaledoniensis* | 1 | 1/(0/0011) | 1.04(0.24, 1.84) | - | - | - | - |

*Confidence Internal.

national control programs, tuberculosis (TB) is still among the highest health hazard in Iran. Owing to the clinical similarity of Nocardiosis to many other infections, TB in particular, *Nocardia* infections are commonly missed/ or not suspected and delay in diagnosis [33]. The preliminary and rapid method for suspected lung infections is acid fast staining on prepared sputum smears for the screening of acid fast bacilli (AFB), however this method alone is unable to differentiate *M. tuberculosis* from non-tuberculous mycobacteria (NTM) and *Nocardia* spp.

Thus, failure in the characterization of *Nocardia* lung infections that are positive in acid fast staining, contributes to the misclassification of these infections, thereby leading to a failure in the treatment of pulmonary Nocardiosis [34]. Since *Nocardia* spp. is abundant in different areas of the country, including the central and southwest parts, performing nucleic acid amplification tests (NAAT) is routinely required for differentiating between *Nocardia* and other AFB smear-positive sputum samples.

In our study, *N. asteroides* was the most frequent isolate species, which supports the findings of other research works [3, 35]. The pathological importance of *Nocardia* infections lies on the pathogenesis process in human body. The pathogenesis mechanism of *Nocardia* related to its ability to survive and grow in a variety of human cells including phagocytic cells by mechanisms including production of catalase and superoxide dismutase and inhibition of phagosome-lysosome fusion [36]. In brief, after the organism enters the body, reticuloendothelial system represents the initial response by mobilization of neutrophils, leads to limiting the dissemination of infection. Later, by the action of cell-mediated immunity macrophage activation occurs, i.e. T-cell population, giving rise to direct lymphocyte-mediated toxicity to the organism. The interplay between phagocytic cells and *Nocardia* hinges on the virulence of the strain

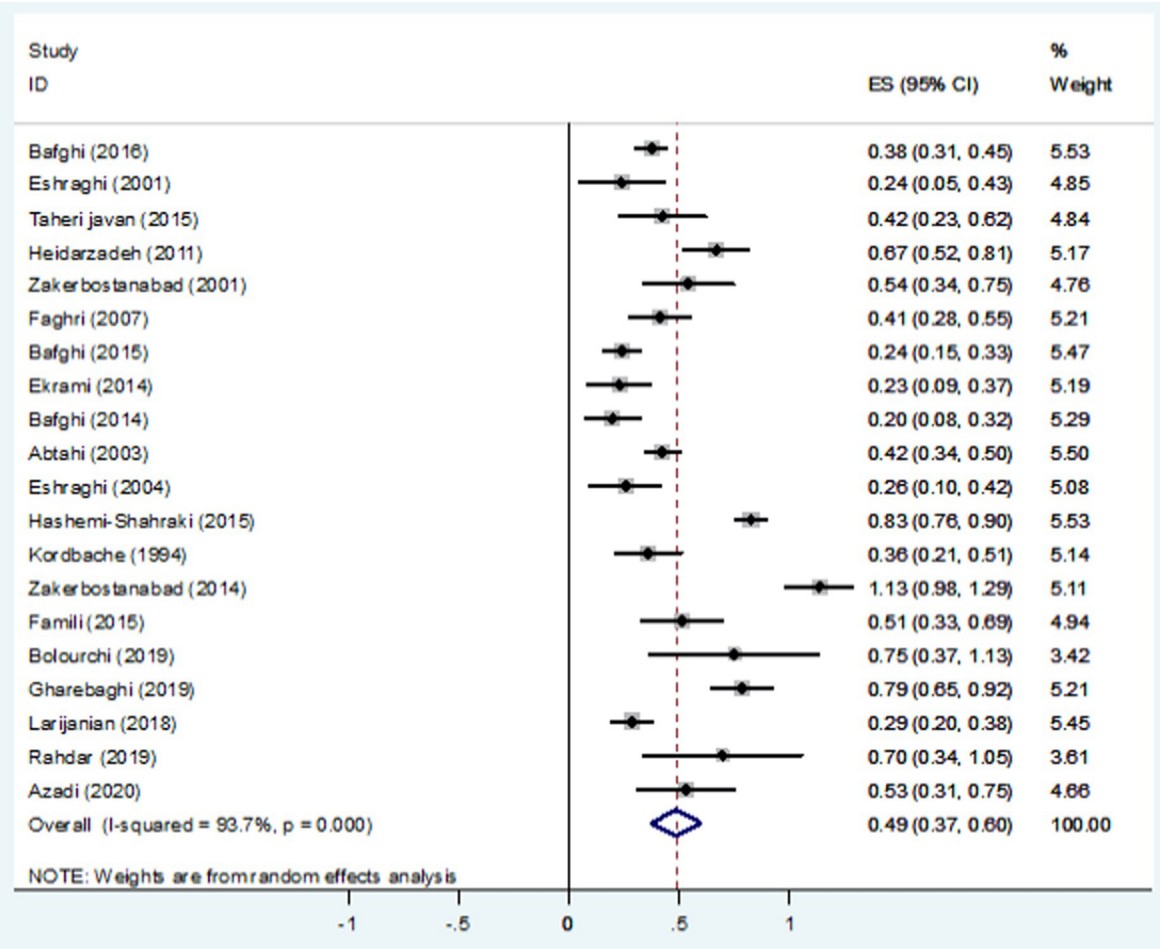

**Fig 2. Forest plot of meta-analysis of *Nocardia* prevalence in Iran based on random-effect models.**

and the growth phase of the nocardial cells. Virulent *Nocardia* can be explained by the complexity of cell wall glycolipids that prevents the fusion of phagosome-lysosome, declines the activity of lysosomal enzyme in macrophages, neutralizes phagosomal acidification, and even withstand the oxidative killing mechanisms of phagocytes. In chronic granulomatous disease, neutrophils and macrophages are unable to produce a burst of oxidative metabolism in the course of phagocytosis, which induces and impairment in the intracellular killing of catalase-positive bacteria viz *Nocardia* species [37]. Ultimately, the host have to enhance a lymphocyte response and then release antibody and/or lymphocyte signals, allowing the phagocytic cells to destroy *N. asteroides*. Pulmonary nocardiosis is the most frequent clinical symptoms of infection because the main route of bacterial exposure is inhalation [37, 38]. In some occasions, the gastrointestinal tract, especially the appendix, is penetrated. In rare conditions, pulmonary infection leads to a dental or periodontal infection. Patients on immunosuppressive drugs, as well as cases with chronic granulomatous disease, chronic alcoholism, diabetes mellitus, and human immunodeficiency virus infection are also more susceptible to pulmonary infections with *Nocardia* [39]. The commencement of symptoms might be acute, subacute, or chronic, and pulmonary nocardiosis, if untreated, can have multiple attributes to the same as

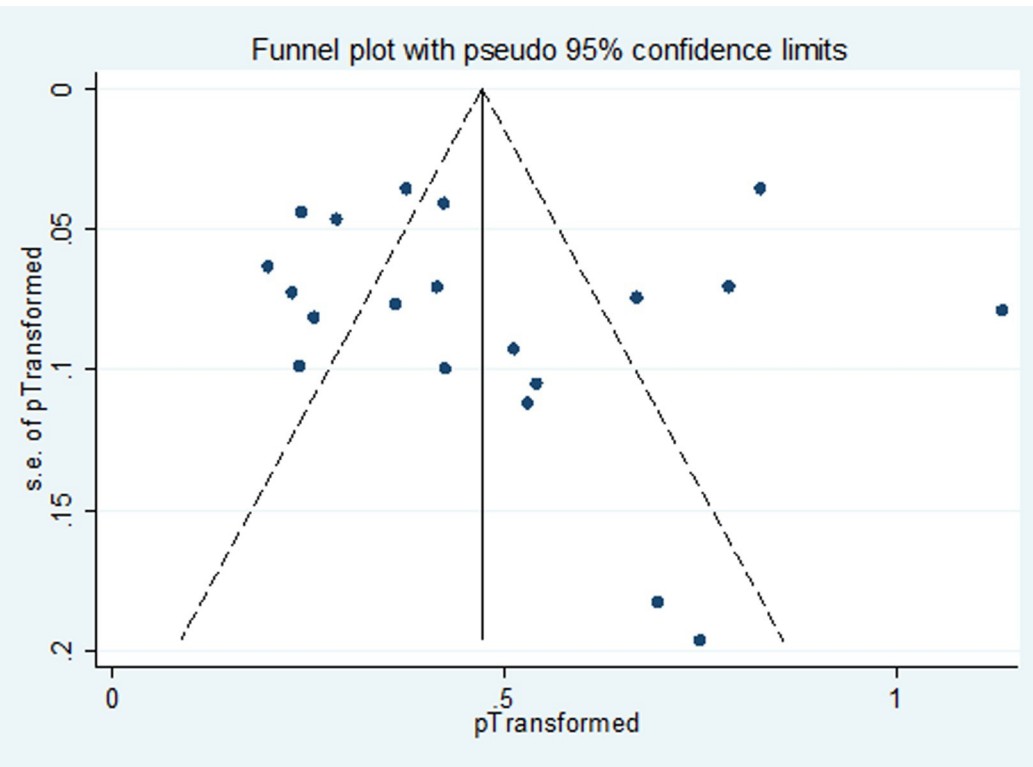

**Fig 3. Funnel plot of the meta-analysis on prevalence of *Nocardia* species.**

tuberculosis, comprising fever, weight loss, nonproductive cough, anorexia, night sweats, dyspnea, hemoptysis. Acute forms of the disease are peculiarly observed in the compromised host [37].

A vast majority of surveys have been performed in the central and southwest regions, but not northern part, of Iran. These data denote that Tehran, the capital city of Iran with many healthcare centers, has a referral role for the whole areas of the country. Therefore, patients, especially those with complicated conditions, are referred to Tehran from all over the country for better management. Considering these data, it seems that the main reason for the isolation of most *Nocardia* spp. in the central provinces is accessibility of the commercial methodologies for detecting this species. Thus, conducting continuous DNA sequencing of homologous genes with a maximum resolution is considerably recommended for areas having a high incidence of *Nocardia* spp. [40].

For the ultimate bacteriological diagnosis of nocardiosis, it is required to isolate and identify the agent from clinical material and from the laboratory where the samples were analyzed [41]. The common diagnostic strategies for *Nocardia* spp. are mostly conventional and molecular methods [42]. In culture media, the growth of Nocardia spp. is slow, and at least two-week incubation is needed. Initial culture discontinuation will diminish the susceptibility of recovery and even may understate the real prevalence of Nocardiosis [37]. Consequently, in view of the clinical presentation and imaging of nocardiosis, which is typically comparable with TB, numerous recent investigations uncovered that the speciation of *Nocardia* may need to be affirmed by molecular methods. So, different molecular techniques have been put forward to precisely identify *Nocardia* spp. In Iran, and a large number of studies have utilized molecular analysis to identify *Nocardia* spp. [10, 12, 13, 18, 19, 24–26, 28, 29]. In cited surveys conducted

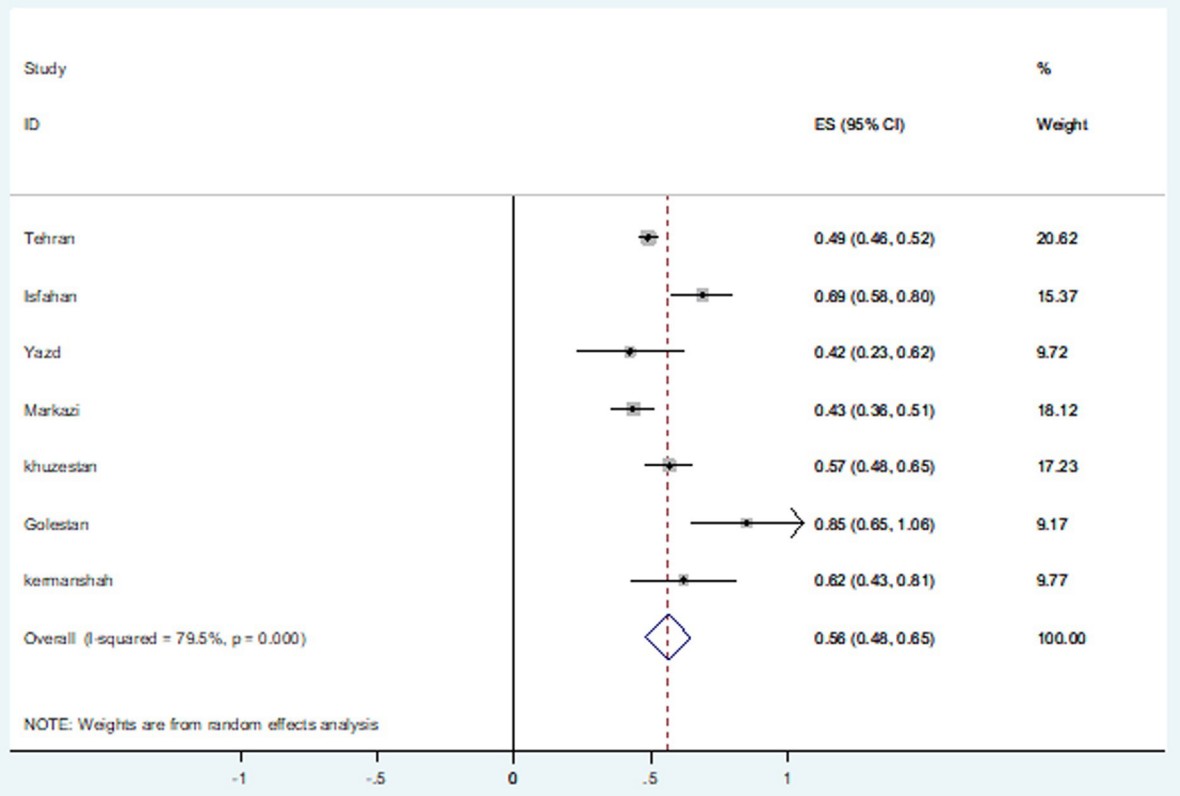

**Fig 4. Forest chart of meta-analysis of *Nocardia* prevalence by provinces of Iran based on random effects model.**

**Table 3. The prevalence of *Nocardia* species in certain provinces of Iran (n = 338).**

| Kermanshah | Tehran | Khuzestan |
|---|---|---|
| *N.asteroides* (2) | Nocardia.spp (28) | Nocardia.spp (1) |
| *N.otitidiscaviarum.caviae* (1) | *N.asteroides* (27) | *N.asteroides* (1) |
| *N.cyriacigeorgica* (1) | *N.otitidiscaviarum.caviae* (34) | *N.otitidiscaviarum.caviae* (1) |
| *N.farcinica* (4) | *N.cyriacigeorgica* (46) | *N.cyriacigeorgica* (1) |
| *N.abscessus* (1) | *N.transvalensis* (1) | *N.farcinica* (1) |
| **Golestan** | *N.farcinica* (17) | *N.carnea* (1) |
| *N.asteroides* (3) | *N.nova* (7) | *N.abscessus* (1) |
| *N.otitidiscaviarum.caviae* (1) | *N.carnea* (3) | *N.wallacei* (1) |
| *N.cyriacigeorgica* (3) | *N.kruczakiae* (2) | **Isfahan** |
| *N.farcinica* (4) | *N.abscessus* (24) | Nocardia.spp (1) |
| *N.abscessus* (2) | *N.a.complex* (5) | *N.asteroides* (1) |
| **Yazd** | *N.veterana* (2) | *N.otitidiscaviarum.caviae* (1) |
| Nocardia.spp (4) | *N.wallacei* (6) | *N.cyriacigeorgica* (1) |
| **Markazi** | *N.arthritidis* (3) | *N.farcinica* (1) |
| *N.asteroides* (26) | *N.brasiliensis* (2) | *N.carnea* (1) |
| *N.cyriacigeorgica* (1) | *N.coubleae* (1) | *N.abscessus* (1) |
| *N.farcinica* (1) | *N.cummidelens* (1) | *N.wallacei* (1) |
| *N.mexicana* (2) | *N.ignorata* (1) | |
| *N.neocaledoniensis* (1) | | |

**Table 4. The distribution of pulmonary Nocardiosis specimens.**

| sputum | polmonary | BAL |
|---|---|---|
| *N. cyriacigeorgica* (26) | | *N. cyriacigeorgica* (30) |
| *N. otitidiscaviarum.caviae* (17) | | *N. otitidiscaviarum.caviae*(17) |
| *N. asteroids* (20) | | *N. transvalensis* (1) |
| *N. coubleae* (1) | | *N. asteroides complex*(3) |
| *N. cummidelens* (1) | | *Nocardiaspp.*(18) |
| *N.ignorata*(1) | | *N. asteroides* (28) |
| *N. asteroides complex* (2) | | - |
| *N. kruczakiae* (1) | | *N. farcinica* (17) |
| *N. carnea* (3) | | *N. wallacei*(8) |
| *N. farcinica* (21) | | *N. carnea*(2) |
| *Nocardiaspp.*(19) | | *N. abscessus*(16) |
| *N. veteran*(1) | | *N. arthritidis*(1) |
| *N.nova*(1) | | *N. kruczakiae* (3) |
| *N. wallacei* (6) | | *N.nova* (4) |

BAL: Broncho Alveolar Lavage.

before years 2000, the identification system was on the basis of biochemical tests and culture, while after this year, PCR was the method that could differentiate recently identified *Nocardia* spp. Since the first step in controlling of the spread of *Nocardia* spp. and the related infections, is precise detection of species by employing sophisticated laboratory methods, so evidence has reflected that the identification of almost all *Nocardia* isolates were carried out by molecular analysis based on the interconnected *gyr*B-16S rRNA gene sequences; thus, PCR remains the gold standard in this regard (Table 6). Besides, though the outbreak of Nocardiosis outbreak is still rare in Iran, however, hospitals should maintain strong infection control practices to avoid outbreaks of Nocardiosis.

In a systematic review, the limitations related to possible publication bias should be taken into consideration. Likewise, the present study has its own shortcomings. First, the Nocardia spp. prevalence among the Iranian population cannot be fully represented as the magnitude of this species has not yet studied in many areas of the country. Second, the probable influence of age, sex, and immigration could not be examined due to the limitation of information achieved from the studied articles. Third, although the number of articles studied was enormous, after several screenings, there were quite small eligible studies. This limitation in the number of

**Table 5. The distribution of extrapulmonary Nocardiosis specimens.**

| | | Extra pulmonary | | |
|---|---|---|---|---|
| wound | Absecus | blood | pleural | skin |
| *N. nova* (2) | *N. cyriacigeorgica* (5) | *N. otitidiscaviarum.caviae* (2) | *N. asteroids* (1) | *N. asteroids* (23) |
| *N. neocaledoniensis* (1) | *N. mexicana*(1) | *N. farcinica* (1) | | *N. abscessus* (5) |
| *N. Mexicana* (1) | *N. farcinica* (3) | *N. cyriacigeorgica* (1) | | *Nocardia spp.*(1) |
| | *N. asteroides* (2) | *Nocardiaspp.*(4) | | *N. otitidiscaviarum.caviae* (1) |
| | *N. abscessus*(12) | | | |
| | *N. carnea* (1) | | | |
| | *N. otitidiscaviarum.caviae* (3) | | | |
| 4 | 26 | 8 | 1 | 30 |

**Table 6. Meta-analysis of the prevalence of *Nocardia* in Iran.**

| Egger test P-Value | Egger test t | Heterogeneity test ‹P-Value | Heterogeneity test ‹I2(%) | Prevalence of nocardia (95% CI) | N. of studies | studies |
|---|---|---|---|---|---|---|
| 0.865 | 0.17 | <0.001 | 93.7 | 0.49(0.37, 0.60) | 20 | All studies |
| 0.493 | -0.75 | 0.154 | 37.8 | 0.38(0.30, 0.45) | 6 | Studies before 2010 |

articles could lessen the statistical power for the detection of funnel plot asymmetry. Fourth, in some studies, detection of Nocardia isolates in many cities of Iran are limited to the application of phenotypic methods, while the use of molecular techniques is more sensitive and specific than conventional methods for diagnosis of Nocardia spp. Fifth, in three studies, the source of samples (pulmonary or extra pulmonary) was not known; therefore, we could not report the exact rate based on sample source.

## Conclusions

In overall, our study presents that despite the fact that *Nocardia* spp. are normally are saprophytic organisms, are currently accounts as emerging pathogens due to an increase in immunocompromised patients among Iranian populations. Therefore, the findings of the present survey could help the programmatic management of the disease within the context of Nocardiosis control programs. Moreover, this review emphasizes on the Nocardia spp. as neglected pathogens and related infections should be takes into account more seriously in future. The distinction of Nocardia infection from other pulmonary infections such as tuberculosis has significant practical importance. Considering our results, the establishment of advanced diagnostic facilities for the rapid detection of *Nocardia* spp. are necessary for optimal therapeutic strategies of Nocardia infections in Iran.

## Supporting information

**S1 Checklist. PRISMA 2009 checklist.**
(DOC)

**S1 Data.**
(DOCX)

**S2 Data.**
(XLS)

## Acknowledgments

This work is part of a research project which was approved in Infectious and Tropical Diseases Research Center, Ahvaz Jundishapur University of Medical Sciences, Ahvaz, Iran. We are grateful to research affairs of the university for their encouragement and support.

## Author Contributions

**Data curation:** Mohammad Hashemzadeh, Aram Asareh Zadegan Dezfuli.

**Formal analysis:** Mohammad Hashemzadeh, Aram Asareh Zadegan Dezfuli.

**Funding acquisition:** Mohammad Hashemzadeh.

**Investigation:** Aram Asareh Zadegan Dezfuli.

**Methodology:** Mohammad Savari, Fatemeh Jahangirimehr.

**Project administration:** Mohammad Savari.

**Writing – original draft:** Azar Dokht Khosravi.

**Writing – review & editing:** Azar Dokht Khosravi.

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
