## [Decision Letter · Decision Letter 0]

14 May 2021

PONE-D-21-07360

Reviews article

Genotypic and Phenotypic diversity Nocardia species in Iran: First systematic Review and Meta-Analysis of data accumulated over years (1992-2020)

PLOS ONE

Dear Dr. khosravi,

Thank you for submitting your manuscript to PLOS ONE. After careful consideration, we feel that it has merit but does not fully meet PLOS ONE’s publication criteria as it currently stands. Therefore, we invite you to submit a revised version of the manuscript that addresses the points raised during the review process.

A major revision is needed.

We look forward to receiving your revised manuscript.

Kind regards,

Abdelazeem Mohamed Algammal, Prof, Ph.D

Academic Editor

PLOS ONE

Journal Requirements:

[The authors thank the Department of Microbiology, School of Medicine, Ahvaz Jundishapur University of Medical Sciences, Ahvaz, Iran and Infectious and Tropical Diseases Research Center, Health Research Institute, Ahvaz Jundishapur University of Medical Sciences, Ahvaz, Iran, for financial support (IR.AJUMS.REC.1398.538).]

 [YES - Specify the role(s) played.]

6. Please ensure that you refer to Figure 5 in your text as, if accepted, production will need this reference to link the reader to the figure.

7. We note that Figure 5 in your submission contain map images which may be copyrighted. All PLOS content is published under the Creative Commons Attribution License (CC BY 4.0), which means that the manuscript, images, and Supporting Information files will be freely available online, and any third party is permitted to access, download, copy, distribute, and use these materials in any way, even commercially, with proper attribution. For these reasons, we cannot publish previously copyrighted maps or satellite images created using proprietary data, such as Google software (Google Maps, Street View, and Earth). For more information, see our copyright guidelines: http://journals.plos.org/plosone/s/licenses-and-copyright.

1.    You may seek permission from the original copyright holder of Figure 5 to publish the content specifically under the CC BY 4.0 license. 

8. Please include a copy of Table 4 and 5 which you refer to in your text on page 7.

10. Thank you for submitting the above manuscript to PLOS ONE. During our internal evaluation of the manuscript, we found significant text overlap between your submission and the following previously published works, some of which you are an author.

- https://storage.googleapis.com/plos-corpus-prod/10.1371/journal.pntd.0002573/1/pntd.0002573.pdf?X-Goog-Algorithm=GOOG4-RSA-SHA256&X-Goog-Credential=wombat-sa%40plos-prod.iam.gserviceaccount.com%2F20210514%2Fauto%2Fstorage%2Fgoog4_request&X-Goog-Date=20210514T163734Z&X-Goog-Expires=3600&X-Goog-SignedHeaders=host&X-Goog-Signature=7a819e3dc520f57398bd690bbc02cc3d63414e931d6ed6ce1e2588d4acc0e9d2127ffa246c7b464cfe07870112717cce290e717b1bbd2738efa660f9b9a9a5356f8730a45715763867a438188781c76e2ea84e689d766defdba1e80870ddbfae8449db9aef235621ec8d4dec780dafc95e3808059ef1c9ae87ab61d3838db858f93e9108adcd39825ef2a32be5fea14ac05c01f5e5b58ea27a39336b174bd1fd82ffc3cd53ee3a5e6dc1f8254a9d8e663b73334ccdab3bf743d655ddaf74b9ad007ca1f88a70cc6c1d3d9bc7b76fbbdaacd92aefff061be2a60fb872b6d20e5fdf10198a2a4521647ab6ab59962506f0e6cd037a3b35067a735a8467492694b6

- https://www.ajicjournal.org/article/S0196-6553(14)00995-X/fulltext

- https://www.journalofinfection.com/article/S0163-4453(16)30244-4/fulltext

Please revise the manuscript to rephrase the duplicated text, cite your sources, and provide details as to how the current manuscript advances on previous work. Please note that further consideration is dependent on the submission of a manuscript that addresses these concerns about the overlap in text with published work.

Reviewers' comments:

Reviewer's Responses to Questions

**Comments to the Author**

1. Is the manuscript technically sound, and do the data support the conclusions?

Reviewer #1: Partly

Reviewer #2: Partly

2. Has the statistical analysis been performed appropriately and rigorously? 

Reviewer #1: I Don't Know

Reviewer #2: Yes

3. Have the authors made all data underlying the findings in their manuscript fully available?

Reviewer #1: Yes

Reviewer #2: No

4. Is the manuscript presented in an intelligible fashion and written in standard English?

Reviewer #1: Yes

Reviewer #2: Yes

5. Review Comments to the Author

Reviewer #1: The article title and conclusion do not correlate with the findings in the work done. Furthermore, there are gross grammatical errors and wrong species names in the article. Perhaps another journal would be more suitable for this piece of work.

Reviewer #2: The authors determine the prevalence of human nocardia spp. in Iran by using a systematic review and meta-analysis according to the preferred reporting items for systematic reviews and meta-Analyses statement.

It is greatly suggested that the manuscript is accepted after minor revisions. In spite of the scientific value and the medical importance of the investigated pathogen, it lacks the presentation of the overall aspects of this pathogen. I have some comments listed below.

-The title is not representative to the data in the review article and lacks the word of after diversity.

-The review is too short and it is not comprehensive to all ideas relating to the studied subject.

-The authors did not detail the phenotypic methods for identification of the investigated pathogen.

-The authors did not talk about the diversity of the pathogen.

-The authors should detail the pathological importance of the pathogen.

-The authors should show the ways to control such pathogen.

-The authors should detail the diagnostic techniques for the pathogen diseases starting from old to recent. The authors must detail the advantages of each detection methods than others.

-The authors should talk about the pathogenesis of the microorganisms implicated iwith referring to their virulence factors.

-There were some errors in the structure of several sentences.

-The authors must write what each abbreviated word stands for before using the abbreviation for the first time.

-All the family, genus, or species names must be typed italic and its first letter must be capitalized in all the manuscript, but when the microorganism is typed without genus or species, it must be as any normal word without capitalization of its first letter or being italicized.

6. PLOS authors have the option to publish the peer review history of their article (what does this mean?). If published, this will include your full peer review and any attached files.

Reviewer #1: No

Reviewer #2: No

---

## [Author Response · Author response to Decision Letter 0]

19 Jun 2021

Response to Reviewers’ comments

PONE-D-21-07360

Reviews article

Genotypic and Phenotypic diversity Nocardia species in Iran: First systematic Review and Meta-Analysis of data accumulated over years (1992-2020)

PLOS ONE

*Note: all corrections are Red-written in this file and Revised Manuscript file with Track Changes.

A major revision is needed.

• -is done

• -is done

• -is done

We look forward to receiving your revised manuscript.

Kind regards,

Abdelazeem Mohamed Algammal, Prof, Ph.D

Academic Editor

PLOS ONE

Journal Requirements:

 -The entire manuscript was re-checked and re-organized according to the Journal's format.

 -The ORCID ID of corresponding author as (0000-0002-7852-6868) was updated in Editorial manager.

[The authors thank the Department of Microbiology, School of Medicine, Ahvaz Jundishapur University of Medical Sciences, Ahvaz, Iran and Infectious and Tropical Diseases Research Center, Health Research Institute, Ahvaz Jundishapur University of Medical Sciences, Ahvaz, Iran, for financial support (IR.AJUMS.REC.1398.538).]

-The Acknowledgement section was added at the appropriate place in the manuscript as:

"This work is part of a research project which was approved in Infectious and Tropical Diseases Research Center, Ahvaz Jundishapur University of Medical Sciences, Ahvaz, Iran. We are grateful to research affairs of the university for their encouragement and support." (Page 18)

-The funding section was deleted. We will add the grant details in the time of submission of revisions in appropriate place in Editorial Manager.

 [YES - Specify the role(s) played.]

-The funding details is already removed from manuscript file and will included in Editorial manager in the time of online submission of revisions.

-We included the funding statement in cover letter as well for your information.

-We applied the revised title identically in both manuscript file and online submission.

-Ethics statement was placed in the beginning of methods section only as below: 

"The initial proposal of the work was approved by the Institutional Review Board (IRB) and Ethics Committee of the Ahvaz Jundishapur University of Medical Sciences, Iran, and necessary permission was granted for the work (IR.AJUMS.REC.1398.538)."(Page 4, lines 84-87)

6. Please ensure that you refer to Figure 5 in your text as, if accepted, production will need this reference to link the reader to the figure.

- According to revision changes, Figure 5 was removed and the related information was added to Results section in the form of Table 3. (Page 11)________________________________________

7. We note that Figure 5 in your submission contain map images which may be copyrighted. All PLOS content is published under the Creative Commons Attribution License (CC BY 4.0), which means that the manuscript, images, and Supporting Information files will be freely available online, and any third party is permitted to access, download, copy, distribute, and use these materials in any way, even commercially, with proper attribution. For these reasons, we cannot publish previously copyrighted maps or satellite images created using proprietary data, such as Google software (Google Maps, Street View, and Earth). For more information, see our copyright guidelines: http://journals.plos.org/plosone/s/licenses-and-copyright.We require you to either (1) present written permission from the copyright holder to publish these figures specifically under the CC BY 4.0 license, or (2) remove the figures from your submission:

- According to revision changes, Figure 5 was removed

1. You may seek permission from the original copyright holder of Figure 5 to publish the content specifically under the CC BY 4.0 license. 

 Table 5 is the work of one of the authors. It has not been removed from the site with a specific source, but it is Photoshop and painting. We assure you that this table has no problem printing

8. Please include a copy of Table 4 and 5 which you refer to in your text on page 7.

-Tables 4 & 5 were re-organized and are now included in the correct place within the revised manuscript. (Page 12)

-Our manuscript does not have any Supporting Information files

10. Thank you for submitting the above manuscript to PLOS ONE. During our internal evaluation of the manuscript, we found significant text overlap between your submission and the following previously published works, some of which you are an author.

- https://storage.googleapis.com/plos-corpus-prod/10.1371/journal.pntd.0002573/1/pntd.0002573.pdf?X-Goog-Algorithm=GOOG4-RSA-SHA256&X-Goog-Credential=wombat-sa%40plos-prod.iam.gserviceaccount.com%2F20210514%2Fauto%2Fstorage%2Fgoog4_request&X-Goog-Date=20210514T163734Z&X-Goog-Expires=3600&X-Goog-SignedHeaders=host&X-Goog-Signature=7a819e3dc520f57398bd690bbc02cc3d63414e931d6ed6ce1e2588d4acc0e9d2127ffa246c7b464cfe07870112717cce290e717b1bbd2738efa660f9b9a9a5356f8730a45715763867a438188781c76e2ea84e689d766defdba1e80870ddbfae8449db9aef235621ec8d4dec780dafc95e3808059ef1c9ae87ab61d3838db858f93e9108adcd39825ef2a32be5fea14ac05c01f5e5b58ea27a39336b174bd1fd82ffc3cd53ee3a5e6dc1f8254a9d8e663b73334ccdab3bf743d655ddaf74b9ad007ca1f88a70cc6c1d3d9bc7b76fbbdaacd92aefff061be2a60fb872b6d20e5fdf10198a2a4521647ab6ab59962506f0e6cd037a3b35067a735a8467492694b6

- https://www.ajicjournal.org/article/S0196-6553(14)00995-X/fulltext

- https://www.journalofinfection.com/article/S0163-4453(16)30244-4/fulltext

Please revise the manuscript to rephrase the duplicated text, cite your sources, and provide details as to how the current manuscript advances on previous work. Please note that further consideration is dependent on the submission of a manuscript that addresses these concerns about the overlap in text with published work.

-Thank you very much for the opportunity have given us to rewrite the manuscript. Rest assured that we re-checked the entire manuscript phrase-by-phrase.

Reviewers' comments:

Reviewer's Responses to Questions

Comments to the Author

1. Is the manuscript technically sound, and do the data support the conclusions?

Reviewer #1: Partly

Reviewer #2: Partly

2. Has the statistical analysis been performed appropriately and rigorously?

Reviewer #1: I Don't Know

Reviewer #2: Yes

3. Have the authors made all data underlying the findings in their manuscript fully available?

Reviewer #1: Yes

Reviewer #2: No

4. Is the manuscript presented in an intelligible fashion and written in standard English?

Reviewer #1: Yes

Reviewer #2: Yes

5. Review Comments to the Author

Reviewer #1: The article title and conclusion do not correlate with the findings in the work done. Furthermore, there are gross grammatical errors and wrong species names in the article. Perhaps another journal would be more suitable for this piece of work.

-The title has been changed as below:

"Genotypic and Phenotypic prevalence of Nocardia species in Iran: First systematic Review and Meta-Analysis of data accumulated over years 1992-2021"

-The conclusion section is also changed according to the findings

Reviewer #2: The authors determine the prevalence of human nocardia spp. in Iran by using a systematic review and meta-analysis according to the preferred reporting items for systematic reviews and meta-Analyses statement.

It is greatly suggested that the manuscript is accepted after minor revisions. In spite of the scientific value and the medical importance of the investigated pathogen, it lacks the presentation of the overall aspects of this pathogen. I have some comments listed below.

-The title is not representative to the data in the review article and lacks the word of after diversity.

-The review is too short and it is not comprehensive to all ideas relating to the studied subject.

-The title has been changed representing the goal of current work. Besides we were able to extend some topics of the work to make it more comprehensive.

The authors did not detail the phenotypic methods for identification of the investigated pathogen. 

-Details of phenotypic methods and biochemical identification tests were added to methods section. (Page 5-6, line 114-123). Regarding molecular methods performed and mentioned in articles after 2000, we have also discussed in Discussion section. (Page 16, lines 307-312)

-The authors did not talk about the diversity of the pathogen.

-The word "diversity" was removed from the title and the word "prevalence" was added instead. So, we have included articles related to prevalence in all three sections of the results.

" Genotypic and Phenotypic prevalence of Nocardia species in Iran:…."

-The authors should detail the pathological importance of the pathogen.

-The pathological importance was added to the text. In Discussion part and a few references was added accordingly (Page 14 line 263-280)________________________________________ 

-The authors should show the ways to control such pathogen.

-The ways to control Nocardia infections were added to the text in Discussion section (Page 16, lines 308-314)

-The authors should detail the diagnostic techniques for the pathogen diseases starting from old to recent. The authors must detail the advantages of each detection methods than others.

-The statement with details were added to discussion section as below:

" In cited surveys conducted before years 2000, the identification system was on the basis of biochemical tests and culture, while after this year, PCR was the method that could differentiate recently identified Nocardia spp. Since the first step in controlling of the spread of Nocardia spp. and the related infections, is precise detection of species by employing sophisticated laboratory methods, so evidence has reflected that the identification of almost all Nocardia isolates were carried out by molecular analysis based on the interconnected gyrB-16S rRNA gene sequences; thus, PCR remains the gold standard in this regard (Table 6). " (Pages 15 & 16, lines 306-312)

-The authors should talk about the pathogenesis of the microorganisms implicated iwith referring to their virulence factors.

-There were some errors in the structure of several sentences.

-The manuscript has been re-checked in terms of grammar and writing errors.

-The authors must write what each abbreviated word stands for before using the abbreviation for the first time.

-All the family, genus, or species names must be typed italic and its first letter must be capitalized in all the manuscript, but when the microorganism is typed without genus or species, it must be as any normal word without capitalization of its first letter or being italicized.

-Thank you for your help and we made corrections 

6. PLOS authors have the option to publish the peer review history of their article (what does this mean?). If published, this will include your full peer review and any attached files.

Do you want your identity to be public for this peer review? For information about this choice, including consent withdrawal, please see our Privacy Policy.

Reviewer #1: No

Reviewer #2: No

Dear Dr. khosravi,

Thank you for your PLOS Publication Fee Assistance application for (). We acknowledge receipt of your application. As part of our case review process, we will be assessing your application based on the information you provided, your manuscript funding disclosure and your co-author affiliations. We may contact you to request further information to help us determine your eligibility for Publication Fee Assistance.

Information about an applicant’s application for fee assistance will not be disclosed to journal editors nor reviewers. PLOS publication decisions are based solely on editorial criteria.

We hope to be able to inform you of the disposition of your PFA request within 10 business days of your application.

Kind regards,

PFA Team

---

## [Decision Letter · Decision Letter 1]

5 Jul 2021

Reviews article

Genotypic and Phenotypic prevalence of Nocardia species in Iran: First systematic Review and Meta-Analysis of data accumulated over years 1992-202 1

PONE-D-21-07360R1

Dear Dr. Khosravi,

We’re pleased to inform you that your manuscript has been judged scientifically suitable for publication and will be formally accepted for publication once it meets all outstanding technical requirements.

Kind regards,

Abdelazeem Mohamed Algammal, Prof, Ph.D

Academic Editor

PLOS ONE

Additional Editor Comments (optional):

Reviewers' comments:

Reviewer's Responses to Questions

**Comments to the Author**

1. If the authors have adequately addressed your comments raised in a previous round of review and you feel that this manuscript is now acceptable for publication, you may indicate that here to bypass the “Comments to the Author” section, enter your conflict of interest statement in the “Confidential to Editor” section, and submit your "Accept" recommendation.

Reviewer #2: All comments have been addressed

2. Is the manuscript technically sound, and do the data support the conclusions?

Reviewer #2: Yes

3. Has the statistical analysis been performed appropriately and rigorously? 

Reviewer #2: No

4. Have the authors made all data underlying the findings in their manuscript fully available?

Reviewer #2: Yes

5. Is the manuscript presented in an intelligible fashion and written in standard English?

Reviewer #2: Yes

6. Review Comments to the Author

Reviewer #2: Regarding this manuscript, It seems sound, but the title is still not relevant and the authors should talk about the pathogenesis of the microorganisms implicated with referring to their virulence factors.

7. PLOS authors have the option to publish the peer review history of their article (what does this mean?). If published, this will include your full peer review and any attached files.

Reviewer #2: No

---

## [Editor Report · Acceptance letter]

13 Jul 2021

PONE-D-21-07360R1 

Genotypic and Phenotypic prevalence of Nocardia species in Iran: First systematic Review and Meta-Analysis of data accumulated over years 1992-2021 

Dear Dr. Khosravi:

I'm pleased to inform you that your manuscript has been deemed suitable for publication in PLOS ONE. Congratulations! Your manuscript is now with our production department. 

Kind regards, 

on behalf of

Professor Abdelazeem Mohamed Algammal 

Academic Editor

PLOS ONE